# Protein feeding mediates sex pheromone biosynthesis in an insect

Shiyu Gui[1], Boaz Yuval[2], Tobias Engl[3], Yongyue Lu[1], Daifeng Cheng[1]*

[1]Department of Entomology, South China Agricultural University, Guangzhou, China; [2]Department of Entomology, Hebrew University of Jerusalem, Rehovot, Israel; [3]Department of Insect Symbiosis, Max Planck Institute for Chemical Ecology, Jena, Germany

**Abstract** Protein feeding is critical for male reproductive success in many insect species. However, how protein affects the reproduction remains largely unknown. Using *Bactrocera dorsalis* as the study model, we investigated how protein feeding regulated sex pheromone synthesis. We show that protein ingestion is essential for sex pheromone synthesis in male. While protein feeding or deprivation did not affect *Bacillus* abundance, transcriptome analysis revealed that sarcosine dehydrogenase (Sardh) in protein-fed males regulates the biosynthesis of sex phero- mones by increasing glycine and threonine (sex pheromone precursors) contents. RNAi-mediated loss-of-function of Sardh decreases glycine, threonine, and sex pheromone contents and results in decreased mating ability in males. The study links male feeding behavior with discrete patterns of gene expression that plays role in sex pheromone synthesis, which in turn translates to successful copulatory behavior of the males.

## Editor's evaluation

The manuscript describes the effects of dietary yeast and sugars on male Bactrocera dorsalis sex pheromone biosynthesis and other mating-related traits. This is an important study showing that yeast feeding stimulates the production of specific sex pheromones and promotes fly mating ability. The data are solid and will be of interest to the fields of chemical ecology and pest management.

*For correspondence: chengdaifeng@scau.edu.cn

## Introduction

Females in many animal species are 'investment breeders', foraging for reproductive resources during adulthood, which are directed into offspring production (*Stearns, 1989*; *Stephens et al., 2009*; *Stephens et al., 2014*). Intriguingly, males of many species may be categorized in a similar manner (*Soulsbury, 2019*), depending on foraging success to secure copulations and manipulate female behavior, while prioritizing different resources than the females (*Gwynne, 2008*; *Ng et al., 2019*). Indeed, a number of studies on males from various insect groups suggest a strong link between adult foraging and reproductive success (e.g. Lepidoptera [*Boggs, 1981*]; Orthoptera [*Gwynne, 2008*]; Mecoptera [*Sauer et al., 1998*]; Diptera [*Yuval et al., 2007*]).

The dipteran family Tephritidae contains over 4000 species, and almost all tephritid adults need post-teneral carbohydrate and protein nutrition to realize their fitness potential (*Pereira et al., 2013*; *Taylor et al., 2013*). Post-teneral protein feeding affects the reproduction of male tephritid flies in multiple ways, including the following: (1) sexual organ development. For example, the reproductive organs (testes, accessory glands, ejaculatory duct, and apodemes) of *Bactrocera dorsalis* males can significantly enlarge after protein consumption (*Reyes-Hernández et al., 2019*). In *Bactrocera tryoni*, protein feeding accelerates the development of the male copulatory apodemes (*Weldon and Taylor,*

*2011*; *Taylor et al., 2013*; *Reyes-Hernández et al., 2019*); (2) pheromone release, which is a key step in reproduction. In *Ceratitis capitata*, protein feeding increases male intensity of sex pheromone release (*Yuval et al., 2002*). *B. tryoni* males release an incomplete mixture of sex pheromones when deprived of protein (*Weldon and Taylor, 2011*) (3) mating behavior. In many species of Tephritidae, mating competitiveness, mating probability, and mating duration are significantly increased in flies supplied with protein (*Yuval et al., 2002*; *Prabhu et al., 2008*; *Taylor et al., 2013*; *Reyes-Hernández et al., 2019*); (4) sperm storage and egg fertilization. Protein supplementation for males increases the amount of sperm stored by mated females (*Taylor et al., 2013*; *Reyes-Hernández et al., 2019*) and the probability of egg fertilization (*Blay and Yuval, 1997*; *Shelly et al., 2002*; *Yuval et al., 2007*); (5) inhibition of female receptivity. Ejaculates of protein-fed males inhibit the remating behavior of females (*Radhakrishnan and Taylor, 2007*). Conversely, the remating rate in females increases significantly when males lack access to protein (*Yuval et al., 2002*; *Taylor et al., 2013*).

Males of the oriental fruit fly, *B. dorsalis*, one of the most economically important tephritid species, attract females by emitting a pheromone produced by bacteria in the rectal glands (*Ren et al., 2021*). The pheromone has been identified as a cocktail of trimethylpyrazine (TMP) and tetramethylpyrazine (TTMP), and the pathway of sex pheromone biosynthesis has been proposed in a previous study (*Zhang et al., 2019*, *Ren et al., 2021*). Although sex pheromone-producing *Bacillus* in *B. dorsalis* have been identified, the source of the precursors has not been determined. Given that protein feeding has a significant influence on the reproductive performance of *B. dorsalis* males (*Shelly and Edu, 2007*), we hypothesized that protein feeding in *B. dorsalis* mediates sex pheromone biosynthesis by affecting the precursor substance content in the rectum. Accordingly, in this study, we investigated the mechanism by which protein feeding influences the biosynthesis of sex pheromones in *B. dorsalis*. The results indicate that protein feeding is the key factor that controls sex pheromone biosynthesis in the male rectum. Ingested protein supplies the glycine and threonine pathway and provides substrates for sex pheromone production.

## Results

### Protein feeding is required for sex pheromone biosynthesis and successful mating

To determine whether protein feeding is required for reproductive performance, we tested the effects of protein feeding on male survival, rectum width, sex pheromones, and mating ability (*Figure 1A*). As in previous studies (*Orankanok et al., 2013*; *Shelly, 2017*), yeast hydrolysate (YH) was used as the protein source. Since rectal *Bacillus* also need to use glucose to synthesize sex pheromone (*Ren et al., 2021*), we fed male flies three different types of sugar (sucrose, fructose, and glucose) to investigate the influence of glucose feeding or not on sex pheromone synthesis and reproductive performance of male *B. dorsalis* (*Figure 1A*). The results showed that different types of sugars had no effect on survival, rectum width, sex pheromone biosynthesis, and mating ability of mature male (12 d old; *Figure 1B–G*). Though protein feeding also did not affect the survival and rectum width of the males (*Figure 1B and C*), major products with the same retention time were only detected in YH-supplemented male rectums when rectal extracts were analyzed by gas chromatography–mass spectrometry (GC–MS; *Figure 1D*). These products were tentatively identified as TMP and TTMP in all three YH-supplemented cases based on their mass spectra (*Figure 1E and F*). And the mating ability of the males deprived of YH was significantly lower than that of the males fed with YH (*Figure 1G*). These results indicate that protein feeding instead of sugar type can significantly affect the male reproductive performance (sex pheromone synthesis and mating ability). Given that sex pheromones are synthesized by rectal *Bacillus* using glucose and amino acids as precursors (*Ren et al., 2021*), we infer that dietary proteins may influence sex pheromone synthesis and mating ability by influencing the abundance of *Bacillus* in the rectum or the amount of the precursors.

### Precursor amino acids of sex pheromones are affected by protein feeding

To examine the above hypotheses, the absolute abundance and composition of the mature male (12 d old) rectum microbial communities were inferred by 16S rRNA gene quantification and amplicon sequencing (*Supplementary file 1*). The results showed that there is no significant difference for

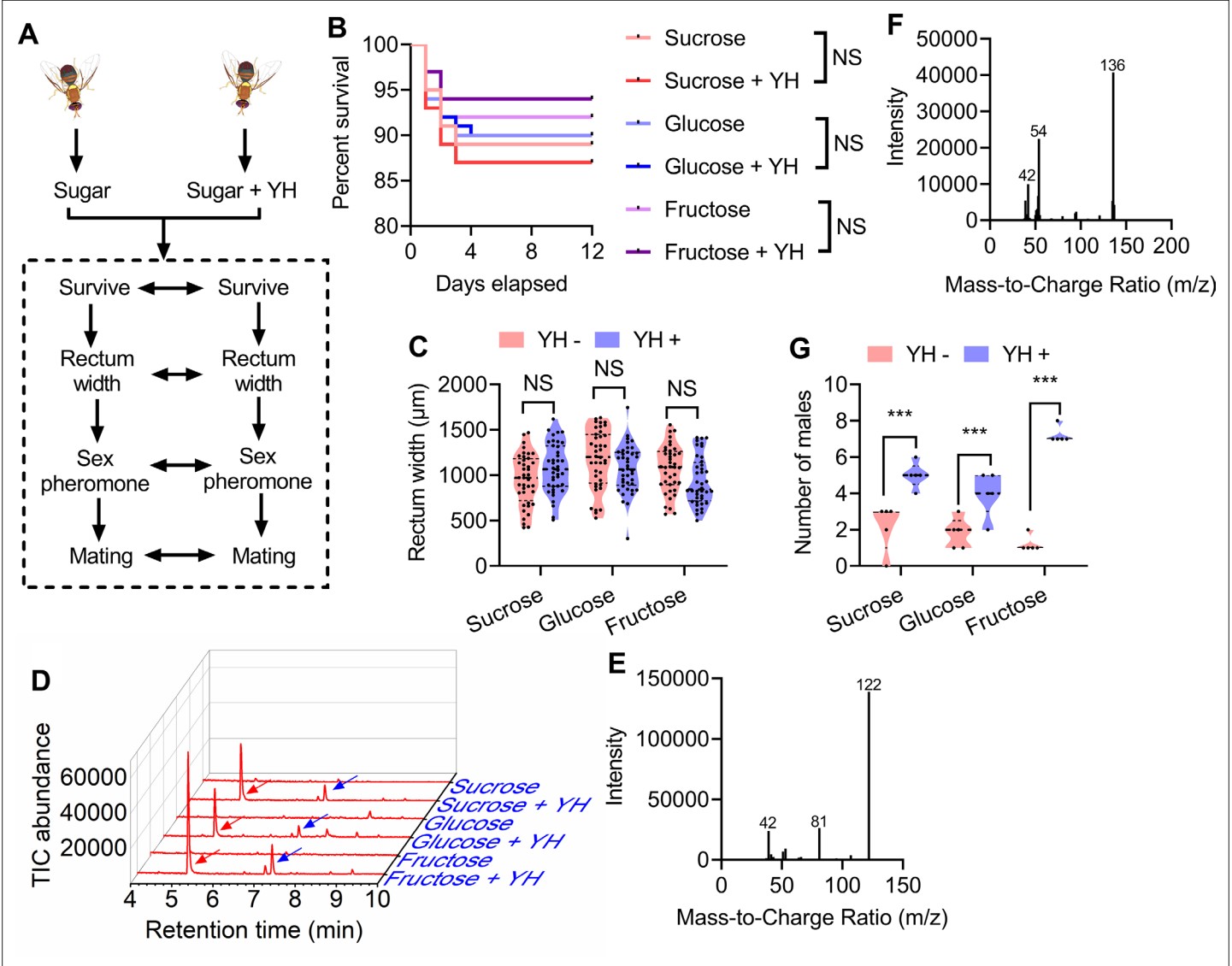

**Figure 1.** Influence of post-teneral protein and sugar feeding on male flies. (**A**) A schematic showing how the male flies were reared and the biological parameters compared. (**B**) Effect of post-teneral protein and sugar on survival (n=200 individuals, Kaplan–Meier survival analysis was used, and NS: no significance). (**C**) Rectum size comparisons between yeast hydrolysate (YH)-deprived (YH−) and YH-fed (YH+) males (n=40 individuals, Student's *t* test, and NS: no significance). (**D**) Gas chromatography–mass spectrometry (GC–MS) ion chromatograms of rectum extracts of males fed different types of food. Traces for the flies fed with YH expressing trimethylpyrazine (TMP; red arrow) and tetramethylpyrazine (TTMP; blue arrow) are shown. (**E**) and (**F**) GC–MS mass spectra of TMP and TTMP. (**G**) Mating ability comparisons between YH− and YH+ males (n=5 replicates, Wilcoxon matched-pairs signed rank test, and *** p<0.001). In violin plots, where the violin encompass the first to the third quartiles, inside the violin the horizontal line shows the median.

The online version of this article includes the following source data for figure 1:

**Source data 1.** Raw data used for analysis in *Figure 1A, B and F*.

total bacteria contents between YH-supplemented and YH-deprived male rectum (*Figure 2A*). 16S rRNA amplicon sequencing results showed that microbial communities at the class level in YH-supplemented male rectum were similar with those in YH-deprived male rectum, especially the abundance of Bacilli was very similar (*Figure 2B and C*). Alpha diversity in 16S rRNA amplicon sequencing also indicated that protein feeding had no influence on diversity except the males feeding on sucrose (*Supplementary file 2*, *Figure 2—figure supplement 1*). These results indicate that protein intake may not affect the abundance of *Bacillus* synthesizing pheromones in the rectum, and sex pheromone loss in YH-deprived males may not be associated with *Bacillus*.

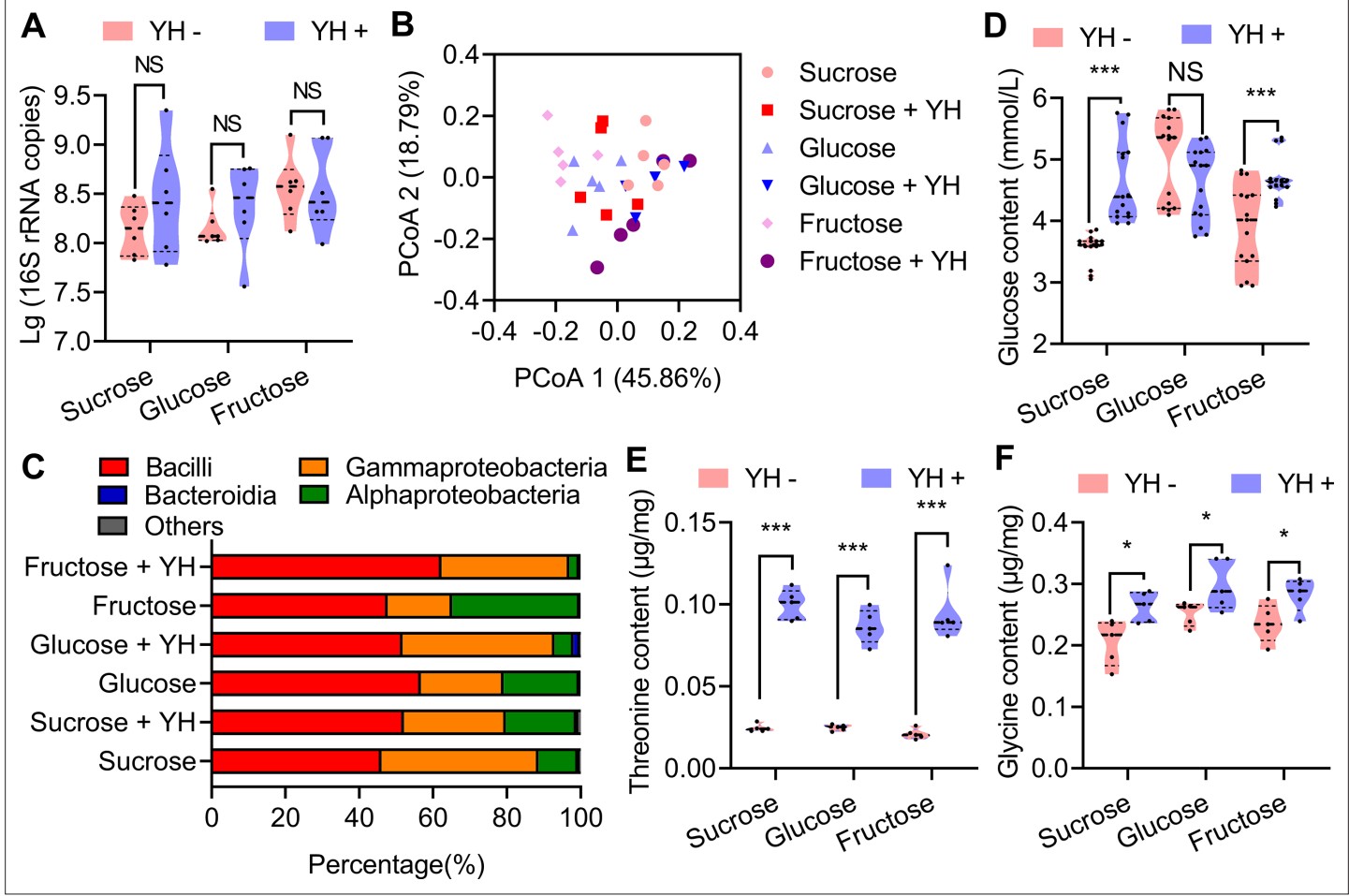

**Figure 2.** Influence of post-teneral protein on rectal bacteria and sex pheromone precursors. (**A**) Boxplot showing total bacteria in the male rectum estimated from 16S rRNA gene quantitative PCR (qPCR; n=6 replicates, Student's *t* test, and NS: no significance). (**B**) Principal coordinate analysis of the microbial community structure (beta diversity and class level) measured by the Bray–Curtis distance matrix of 16S rRNA gene amplicon sequences. (**C**) Class-level relative abundance of 16S rRNA gene amplicon sequences. Values are averaged according to yeast hydrolysate (YH)-deprived and YH-supplied males. (**D**), (**E**), and (**F**) Influence of YH supply on rectum glucose (n=15 replicates), threonine (n=5 replicates), and glycine (n=5 replicates) contents (Student's *t* test, NS: no significance, * $p<0.05$, and *** $p<0.0001$). In violin plots, where the violin encompass the first to the third quartiles, inside the violin the horizontal line shows the median.

The online version of this article includes the following source data and figure supplement(s) for figure 2:

**Source data 1.** Raw data used for analysis in *Figure 2A, D, E and F*.

**Figure supplement 1.** Boxplots showing the estimated diversity of the microbial community based on Shannon (**A**) and Inverse Simpson (**B**) indices of the 16S amplicon sequences (n=5 replicates, Student's *t* test, NS: no significance, * $p<0.05$, and ** $p<0.001$).

Since rectal *Bacillus*, glucose, and threonine/glycine are three essential factors for sex pheromone synthesis, we further measured glucose and threonine/glycine contents to confirm whether protein supplementation regulates sex pheromone synthesis by influencing rectal glucose or threonine/glycine contents. The results showed that YH feeding only increased glucose contents in males in the sucrose and fructose groups (*Figure 2D*), which indicates that increased rectal glucose content caused by protein supplementation may not be the main factor affecting sex pheromones synthesis. On the other hand, YH feeding significantly increased threonine and glycine contents in males in all sugar groups (*Figure 2E and F*), which indicates that the decreased precursor amino acid contents may be the main factor affecting sex pheromone producing in rectum of YH-deprived males.

## Glycine and threonine pathway involved in protein metabolism

If YH supplementation is necessary for threonine and glycine synthesis, we reasoned that molecular pathways in the rectum mediating threonine and glycine synthesis may show different expression

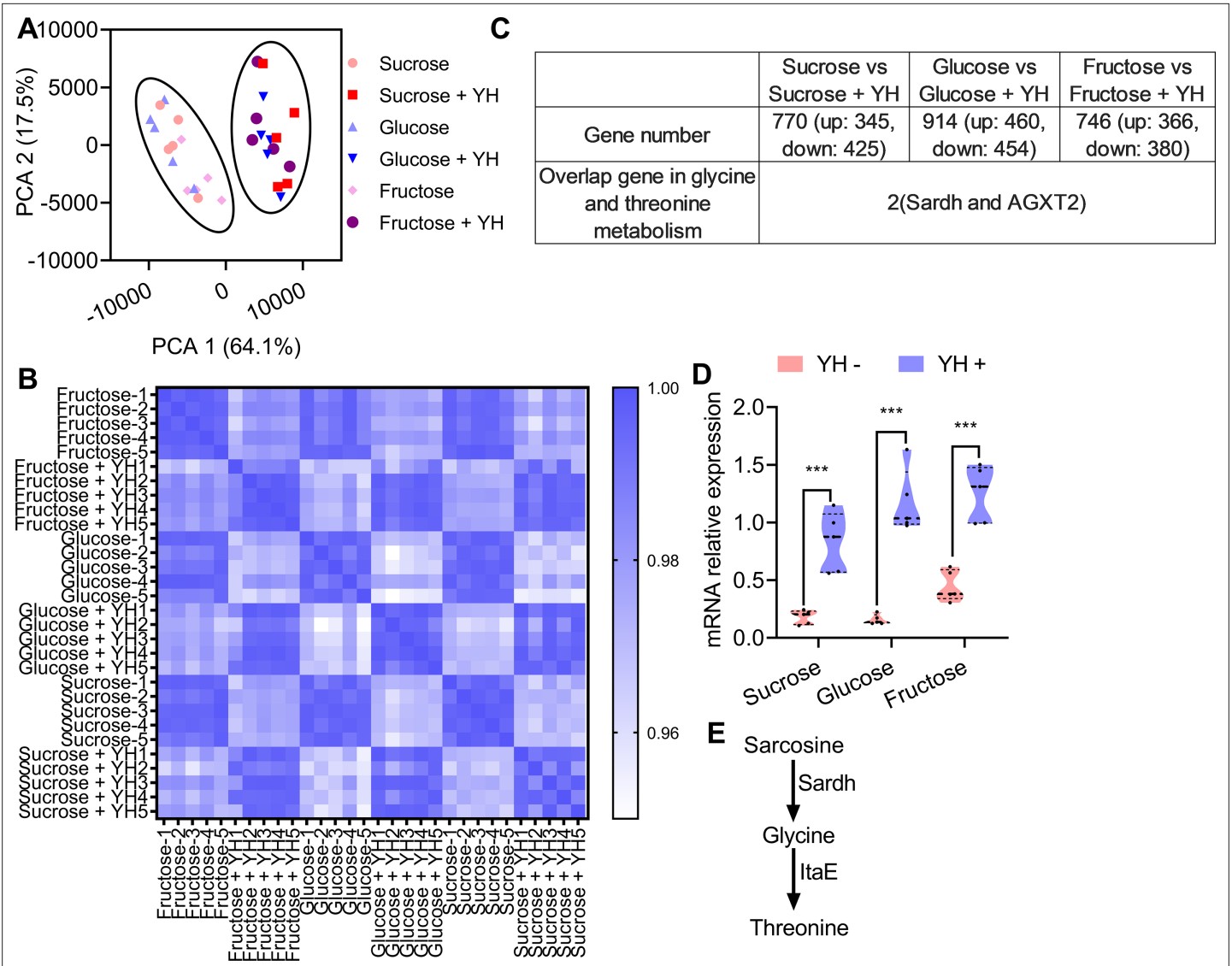

**Figure 3.** Transcriptome comparisons between yeast hydrolysate (YH)-fed and YH-deprived males. (**A**) Principal component analysis (PCA) obtained from gene expression profiles showing differences between YH-fed and YH-deprived males. Flies are clustered according to YH fed or not. (**B**) Pearson correlation coefficient showing the similarity between the samples. Higher similarity of the transcriptome is shown by a darker blue color (higher correlation coefficient). (**C**) Table showing the number of genes found in any given category and the genes involved in the threonine metabolism pathway between comparisons. (**D**) Quantitative PCR (qPCR) verifying the expression of Sardh in YH-fed and YH-deprived males (n=5 replicates, Student's *t* test, and *** p<0.001). In violin plots, where the violin encompass the first to the third quartiles, inside the violin the horizontal line shows the median. (**E**) Proposed model of the threonine metabolism pathway in insects.

The online version of this article includes the following source data and figure supplement(s) for figure 3:

**Source data 1.** Raw data used for analysis in Figure 3C.

**Source data 2.** Raw data used for analysis in *Figure 3A and D*.

**Figure supplement 1.** Differential expressed genes in sucrose (**A**), glucose (**B**), and fructose (**C**) groups.

**Figure supplement 2.** Fragments Kilobase of exon model per millon mapped reads(fpkm) values of the differentially expressed genes in glycine, serine, and threonine metabolism pathway (n=5 replicates, Student's *t* test, and *** p<0.0001).

patterns between YH-fed individuals and YH-deprived individuals. We first carried out RNA-seq in the rectum of YH-fed males and YH-deprived males (12 d old). Principal component analysis (PCA) using the expression profiles of the identified genes indicated that YH-deprived males were significantly different from the YH-fed ones (*Figure 3A*, *Supplementary file 3*). Pearson correlation coefficients, which were generated by the expression profiles, between samples also indicated that YH-fed males

had higher similarity than YH-deprived individuals (*Figure 3B*). Pairwise differential expression (DE) analysis identified 770, 914, and 746 DE genes in the sucrose, glucose, and fructose groups, respectively (*Supplementary file 4*, *Figure 3—figure supplement 1*). To identify the DE genes involved in synthesizing glycine or threonine, a Kyoto Encyclopedia of Genes and Genomes (KEGG) pathway enrichment analysis was performed. The glycine and threonine pathway was significantly enriched in the sucrose, glucose, and fructose groups, with the sarcosine dehydrogenase gene (Sardh) and alanine-glyoxylate transaminase (AGXT2) being the significantly differentially expressed genes (DEGs) in all groups, and the expression of Sardh was significantly induced in YH-fed males (*Figure 3C*, *Figure 3—figure supplement 2*, *Supplementary file 5*). Quantitative PCR (qPCR) also confirmed that Sardh expression was significantly enhanced by YH feeding, yet unaffected by sugar identity (*Figure 3D*). In the glycine and threonine metabolism pathway in insects, Sardh and L-threonine aldolase (ltaE) are responsible for converting sarcosine into glycine (*Frisell and Mackenzie, 1962*) and threonine (*Liu et al., 1998*), respectively (*Figure 3E*). Together, the results suggest that Sardh might be involved in sex pheromone biosynthesis by controlling glycine and threonine synthesis.

In normally reared (sucrose and protein were both provided) male *B. dorsalis*, sex pheromones can only be produced 9 d after emergence (*Ren et al., 2021*). Thus, we measured the threonine and glycine contents and generated RNA-seq datasets of the male rectum from YH and sucrose fed males at different development times (0 d, 3 d, 6 d, 9 d, and 12 d) to further verify whether the amino acid

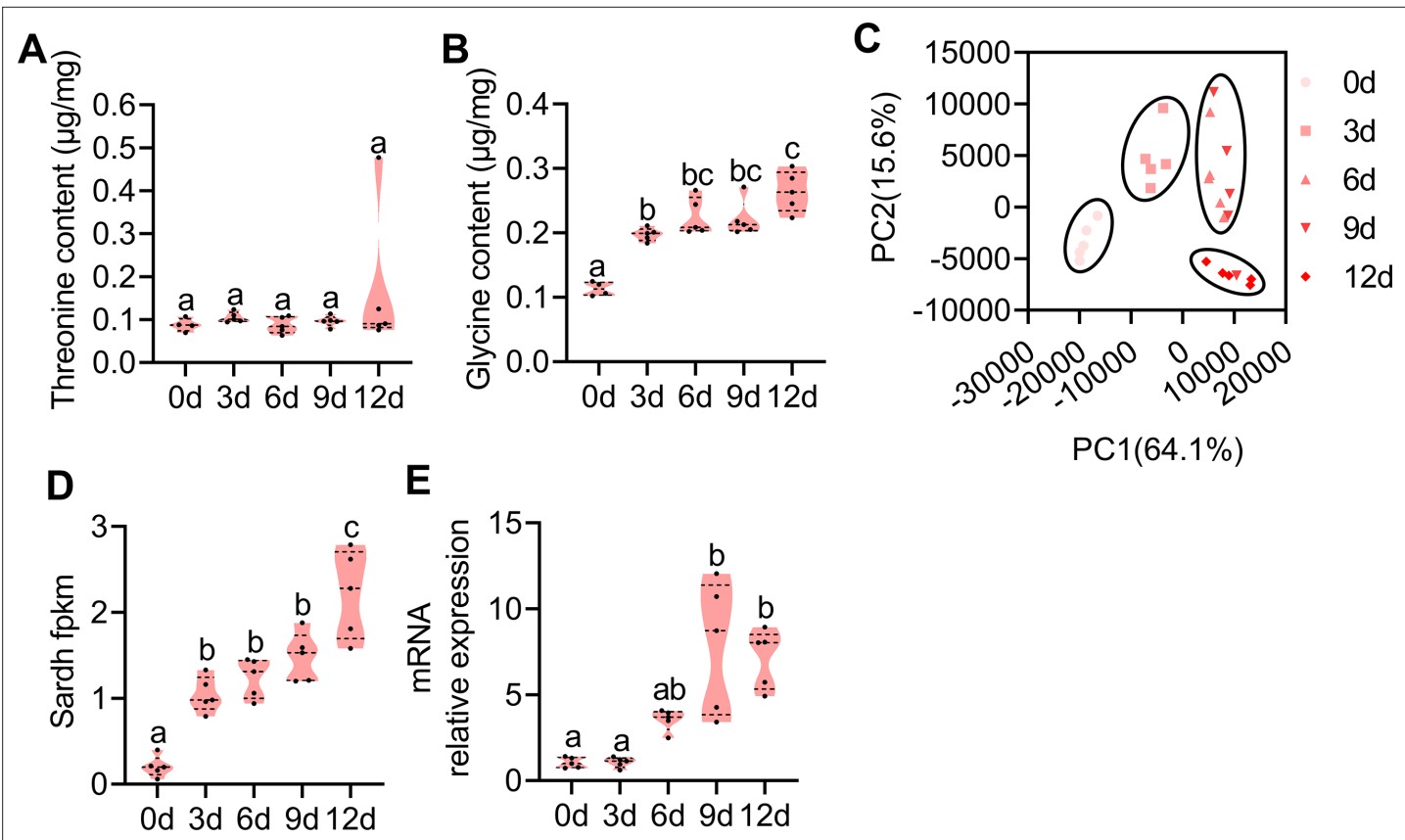

**Figure 4.** Amino acid contents and transcriptome investigation of male rectums at different developmental stages. (**A and B**) Threonine (n=5 replicates) and glycine (n=5 replicates) contents in the rectum at different developmental stages (different letters above the error bars indicate significant differences at the 0.05 level analyzed by ANOVA followed by Tukey's test). (**C**) Principal component analysis (PCA) using differential expression (DE) genes obtained from pairwise comparisons between different developmental stages. (**D and E**) Expression profiles of Sardh obtained by RNA-seq and quantitative PCR (qPCR; n=5 replicates, different letters above the error bars indicate significant differences at the 0.05 level analyzed by ANOVA followed by Tukey's test). In violin plots, where the violin encompass the first to the third quartiles, inside the violin the horizontal line shows the median.

The online version of this article includes the following source data and figure supplement(s) for figure 4:

**Source data 1.** Raw data used for analysis in *Figure 4A, B, D and E*.

**Figure supplement 1.** Transcriptome comparisons between different development stages.

contents and Sardh expression were associated with sex pheromone biosynthesis. Consistent with the idea that the glycine and threonine metabolism pathway was involved in sex pheromone biosynthesis, we found that the glycine content was significantly higher in older males (6 d, 9 d, and 12 d) (although there was no difference in the threonine content; *Figure 4A and B*). RNA-seq data indicated that the older males had higher similarity for gene expression profiles (*Figure 4C*, *Figure 4—figure supplement 1*, *Supplementary file 6*). DEG analysis also indicated that a larger number of DE genes occurred in males with greater age differences (*Figure 4—figure supplement 1*, *Supplementary file 7*). KEGG analysis indicated that more DE genes were enriched in the glycine and threonine pathway between males with greater age differences (*Supplementary file 8*). Consistently, Sardh in the glycine and threonine pathway was significantly highly expressed in the rectum of older males (*Figure 4D and*

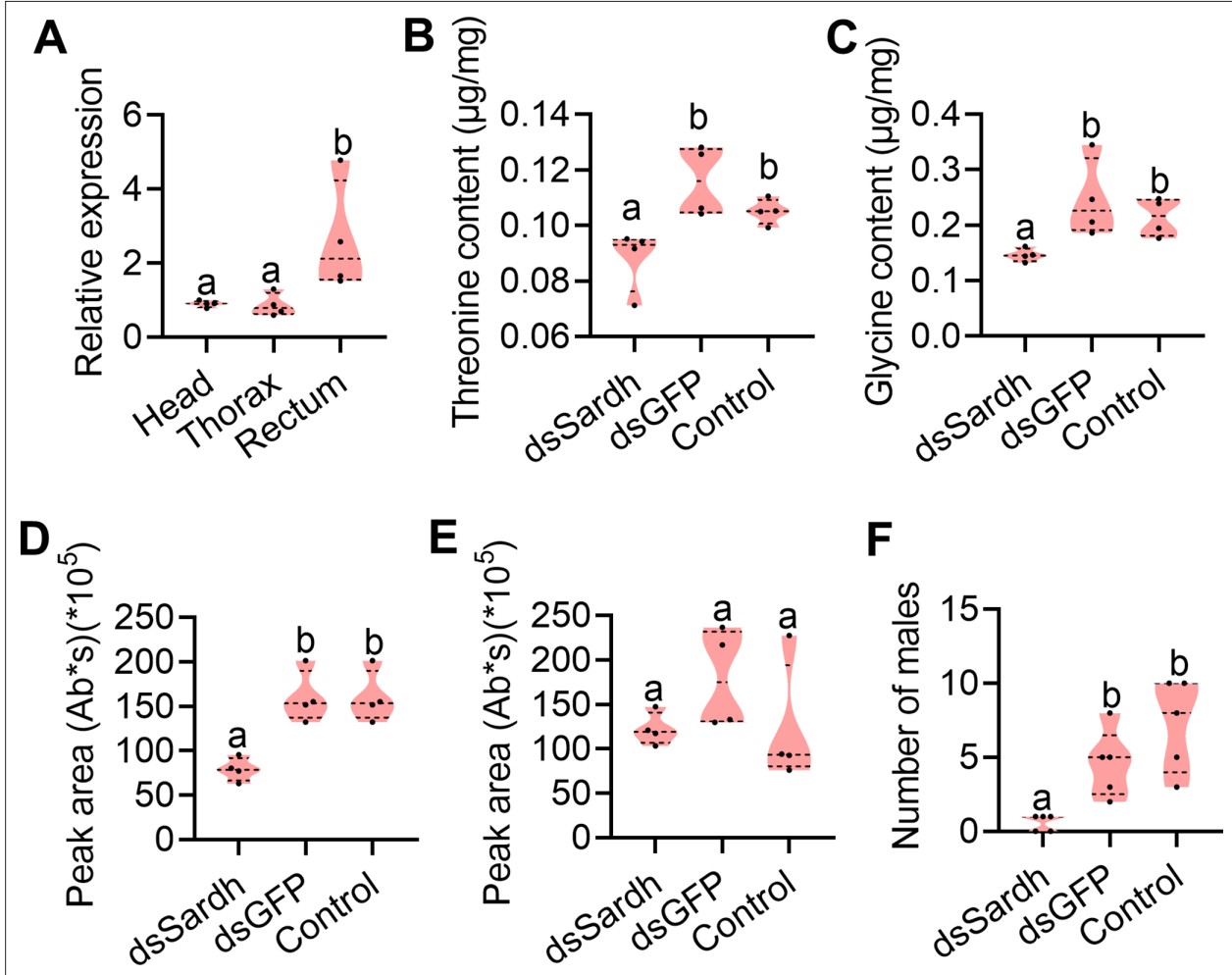

**Figure 5.** Functional verification of Sardh in sex pheromone biosynthesis. (**A**) Expression of Sardh in different tissues with quantitative PCR (qPCR; n=5 replicates, different letters above the error bars indicate significant differences at the 0.05 level analyzed by ANOVA followed by Tukey's test). (**B and C**) Threonine (n=5 replicates) and glycine (n=5 replicates) contents in the rectum with Sardh knockdown (different letters above the error bars indicate significant differences at the 0.05 level analyzed by ANOVA followed by Tukey's test). (**D and E**) Sex pheromone (trimethylpyrazine [TMP] and tetramethylpyrazine [TTMP]) quantification in the rectum with Sardh knockdown (n=4 replicates, different letters above the error bars indicate significant differences at the 0.05 level analyzed by ANOVA followed by Tukey's test). (**F**) Mating ability comparisons between males with Sardh knockdown and controls (n=5 replicates, different letters above the error bars indicate significant differences at the 0.05 level analyzed by Kruskal–Wallis test followed by Dunn's multiple comparisons test). In violin plots, where the violin encompass the first to the third quartiles, inside the violin the horizontal line shows the median.

The online version of this article includes the following source data and figure supplement(s) for figure 5:

**Source data 1.** Raw data used for analysis in *Figure 5A–F*.

**Figure supplement 1.** RNAi efficiency of Sardh after 48 hr (n=3 replicates, different letters above the error bars indicate significant differences at the 0.05 level analyzed by ANOVA followed by Tukey's test).

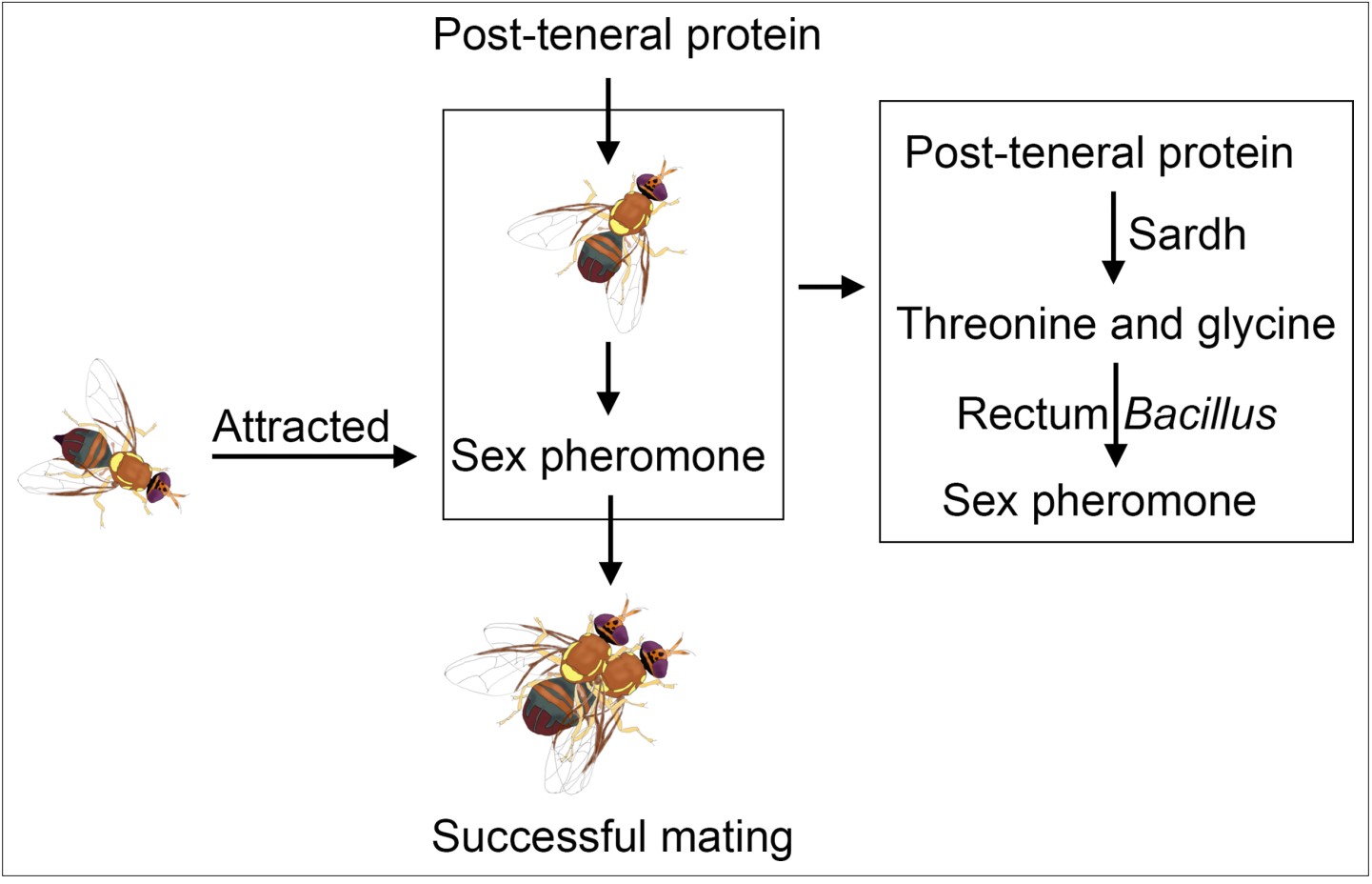

**Figure 6.** Schematic illustrating the sex pheromone biosynthesis hypothesis that *B. dorsalis* rectum. *Bacillus* uses threonine and glycine, which are converted by Sardh with post-teneral protein feeding by *B. dorsalis*, as precursor substances to synthesize the sex pheromone.

*E*). The high expression of Sardh in the rectum suggests that Sardh plays a role in converting YH into glycine and threonine to produce sex pheromones.

### Functional Sardh is necessary for sex pheromone biosynthesis

We next focused on genetically testing whether Sardh was necessary for sex pheromone biosynthesis. To this end, we first measured the relative expression level of Sardh in the head, thorax, and rectum of mature males (12 d old) fed with YH and sucrose to further confirm that Sardh plays a role in sex pheromone biosynthesis in tissue. The qPCR results showed that Sardh was indeed highly expressed in the rectum (*Figure 5A*). We then performed RNAi in Sardh by injecting dsRNA into the male (12 d old) abdomen and checked the influence on precursor contents and reproductive performance (*Figure 5—figure supplement 1*). Similar to YH-deprived males, Sardh knockdown males showed significantly decreased rectal threonine and glycine contents (*Figure 5B and C*). Sex pheromone quantification results indicated that TMP content in the rectum decreased significantly in Sardh knockdown males (*Figure 5D*) and that Sardh knockdown males showed significantly decreased mating competition ability (*Figure 5F*). These results show that Sardh plays role in converting the rectal threonine and glycine. Together, the findings provide a functional demonstration that Sardh, which can be induced by protein feeding and plays a role in synthesizing glycine and threonine, is necessary to regulate sex pheromone biosynthesis in male *B. dorsalis*.

### Discussion

In recent decades, a large number of studies have reported that protein feeding is critical for male insect reproductive success. In the study, how ingested proteins supply the precursors of sex pheromones to

male *B. dorsalis* was indicated. Highly expressed Sardh can convert protein into threonine and glycine, which can be used by the rectum *Bacillus* to synthesize the sex pheromone (*Figure 6*). The study clarifies the molecular mechanism by which host protein feeding regulates sex pheromone biosynthesis.

A large number of studies have indicated that pyrazines are widely used as pheromones in insects (*Bohman et al., 2016*; *Calcagnile et al., 2019*). However, how pyrazines are synthesized in these insects has not yet been revealed. With a series of chemical analysis and molecular biology experiments, we discovered that protein fed by insects contributes to provide precursor substances for pyrazines synthesis. Specifically, we confirmed that the Sardh in the glycine and threonine pathway can convert protein into pyrazine precursor substances-threonine and glycine. Given that the glycine and threonine pathway is conserved in insects (*Crawford et al., 2010*; *Nallu et al., 2018*; *Sonn et al., 2018*), our findings may be relevant for all insects that use protein to synthesize pyrazines.

Previous studies suggest that the influence of protein feeding on the mating success of Tephritidae is caused by affecting the development of testes and accessory glands (*Weldon and Taylor, 2011*; *Taylor et al., 2013*; *Reyes-Hernández et al., 2019*) and increased levels of courtship activity (*Pereira et al., 2013*). Although researchers have speculated that there is a positive correlation between protein feeding and sex pheromones (*Yuval et al., 2007*), this relationship has been hard to pin down. Certain plant chemicals, such as methyl eugenol, gingerone, and raspberry ketone, which strongly attract tephritidae males of some species, are thought to be the precursors of sex pheromones (*Tan and Nishida, 2012*; *Kumaran et al., 2014a*, *Kumaran et al., 2014b*, *Segura et al., 2018*), and a variety of chemicals have been identified and proposed as sex pheromone components in fruit flies (*Chuman et al., 1987*; *Baker and Heath, 1993*; *Wicker-Thomas, 2007*; *Robacker et al., 2009*; *Hiap et al., 2019*; *Levi-Zada et al., 2020*; *Ono et al., 2020*). However, the biosynthetic pathways of only some suspected pheromones have been elucidated. We have proposed here that protein ingested by *B. dorsalis* is converted into threonine and glycine, which are precursor substances of the sex pheromone. The positive relationship between protein feeding and reproductive performance in Tephritidae is elucidated in this case by showing that proteins play a role in supplying precursor substances for sex pheromone biosynthesis. One thing should be noted is that glycine and threonine levels may be elevated partly because the flies fed on YH. To answer such question, we need to further determine the composition and content of amino acids in the YH. We need to determine if the YH contains large amounts of glycine and threonine that can enter into the rectum and be used by the rectal *Bacillus*. In addition to investigating the influence of protein on sex pheromones, the roles of sugars were also tested by feeding males different types of sugars. The results indicate that the amount of sex pheromone produced is significantly affected by the type of sugar (*Figure 1D*). Males fed fructose produced much higher amounts of sex pheromones than glucose-fed males. However, glucose has been described as the precursor substance for generating TMP and TTMP. We speculate that fructose may also be used in the sex pheromone production process, and the utilization efficiency of fructose may be much higher. In glycolysis, glucose is first converted into fructose 6-phosphate and then regenerated into pyruvate, which is involved in the synthesis of TMP and TTMP (*Xiao et al., 2014*; *Xu et al., 2018*; *Zhang et al., 2019*). However, fructose can be catalyzed directly by hexokinase to form fructose 6-phosphate in the glycolysis process. Such a step can omit the step of glucose to fructose 6-phosphate, which may increase pyruvate conversion efficiency and then generate more TMP and TTMP. This may also be related to the fact that *B. dorsalis* has a preference for hosts that are fructose-rich fruits. As males congregate on these hosts to attract mates, opportunities for feeding on fruit juices and fruit exudates may abound.

Previous studies have shown that in the glycine and threonine pathway of insects and bacteria, Sardh converts sarcosine into glycine (*Frisell and Mackenzie, 1962*), and ltaE converts glycine into threonine (*Liu et al., 1998*). However, how protein ingested by *B. dorsalis* affects the production of sarcosine remains to be investigated. Does protein supplementation provide sarcosine directly to the fly? Or does protein supplementation provide the fly with other substances that can be converted into sarcosine? If so, what are these substances? How are they converted to sarcosine? These questions are very complex, and more experimental evidence is needed to uncover them. Both glycine and threonine were elevated in the YH-fed flies. However, only Sardh (that converts sarcosine to glycine) was upregulated and ltaE (that converts glycine to threonine) was not differentially expressed. Does YH provide threonine directly to the flies, or are there other ways to synthesize threonine? These questions also need to be confirmed by further experiments. Moreover, we found that Sardh knockdown

males showed significantly decreased rectal threonine and glycine contents, but TTMP level was not significantly reduced (*Figure 5E*). In the pyrazine synthesis pathway of *Bacillus*, two molecules of glucose can be converted into TTMP, while one molecule of glucose and one molecule of threonine (or glycine) can be converted into TMP (*Zhang et al., 2019*). Therefore, we speculate that the reason why TTMP level is not affected is that glucose content in the rectum is not regulated by Sardh. Nevertheless, the study links male feeding behavior with discrete patterns of gene expression that lead to pheromone production.

## Materials and methods

### Insect rearing

The *B. dorsalis* strain collected from a *carambola* (*Averrhoa carambola*) orchard in Guangzhou, Guangdong Province, was reared under laboratory conditions (27 ± 1°C, 12:12 hr light:dark cycle, 70–80% RH(Relative humidity)). A maize-based artificial diet containing 150 g of corn flour, 150 g of banana, 0.6 g of sodium benzoate, 30 g of yeast, 30 g of sucrose, 30 g of paper towel, 1.2 ml of hydrochloric acid, and 300 ml of water was used to feed the larvae. Adults were fed a solid diet (consisting of 50 g YH [protein food] and 50 g sugar) and 50 ml sterile water. Approximately, 200 adults were held in a 35 cm × 35 cm × 35 cm wooden cage. To test the effect of protein supplementation on sex pheromone synthesis, flies fed only sugar and sterile water were also prepared. To test the effect of different type of sugar on sex pheromone synthesis, flies fed with sucrose, glucose, or fructose as sugar were also prepared.

### Sex pheromone identification in the *B. dorsalis* rectum

60 rectums of 12 d old males fed different types of diets were dissected at 20:00 P.M. Sex pheromones in the rectum were extracted with 500 µl n-hexane by shaking (180 rpm) in a 30°C incubator for 24 hr. Then, GC–MS with an Agilent 7890B Series GC system coupled to a quadrupole-type-mass-selective detector (Agilent 5977B; transfer line temperature: 230°C, source temperature: 230°C, and ionization potential: 70 eV) was used to identify sex pheromones in the rectum extraction according to our previous method (*Ren et al., 2021*).

### Effect of protein feeding on biological parameters

Flies that were fed with and without YH were prepared to determine the effect of protein on biological parameters (adult survival, rectum width, rectum glucose content, rectum threonine content, rectum glycine content, and mating ability). To study survival, the studies were initiated with six groups of newly emerged males (200 males). Each group was maintained separately and was provided different types of food (sucrose, sucrose + YH, glucose, glucose + YH, fructose, and fructose +YH). The mortality of the males was recorded each day until the males matured (12 d later). The rectum width of the mature males was measured.

### Glucose content measurement

The glucose content in the rectum of the mature males was measured with a glucometer. To determine glucose content, the rectums of 12 mature males were collected and placed in a 1.5 ml microcentrifuge tube containing 10 µl of sterile Milli-Q water. Then, the samples were ground with a grinding machine. The samples were centrifuged for 15 min at 12,000 rpm. Then, the supernatants were collected and analyzed with a glucometer (ONETOUCH, Verio Flex). Then, glucose contents were normalized to rectum weight and compared between different treatments.

### Amino acid content measurement

For threonine and glycine identification, sample preparation for free amino acid analysis was performed as described by *Shahzad et al., 2019*. Briefly, the rectums of 15 mature males were collected and placed in a 1.5 ml microcentrifuge tube containing 500 µl of sulfosalicylic acid solution (5%, diluted in water). Then, the samples were ground with a grinding machine. The samples were centrifuged for 15 min at 12,000 rpm. Then, the supernatants were collected in another centrifuge tube, and 1 ml of sulfosalicylic acid solution was added. Then, the threonine content in the samples was quantified

with an amino acid analyzer (Hitachi L-8900, Japan) according to the standard method. Then glycine/threonine contents were normalized to rectum weight and compared between different treatments.

## Mating competition assays

Mating competition between YH-deprived males and control males was performed in a 35 cm × 35 cm × 35 cm wooden cage. Briefly, 60 mature males with colored pronota (30 YH-deprived males [red] and 30 control males [green]) were placed in one cage, and then 30 mature unmated females were placed in the cage at 8:00 P.M. Mating behavior was observed for 2 hr, and the number of mated males was recorded and compared. Five replicates were conducted for each diet pair.

## Effect of protein feeding on rectal bacterial diversity

To analyze bacterial diversity in the male rectum, the rectums of five males fed different foods were collected (five replicate samples were prepared). Then, bacterial DNA was extracted from the rectum samples using the Bacterial Genomic DNA Extraction Kit (Tiangen, Beijing, China) according to the manufacturer's protocol. qPCR (16S-338F and 16S-518R primers were used [*Supplementary file 9*]) was used to estimate the absolute abundance of bacteria in the rectum according to our previous method (*Ren et al., 2021*). The 16S rRNA V3–V4 region was amplified by PCR (16S-341F and 16S-806R primers were used [*Supplementary file 9*]). Then, the amplicons were purified and sequenced (2×250) on an Illumina HiSeq 2500 platform. The software Mothur was used to cluster tags of more than 97% identity into OTUs(Operational Taxonomic Unit), and then the abundances of the OTUs were calculated. The taxonomic classification of OTUs was based on the annotation result of contained tags according to the mode principle; that is, the taxonomic rank that contained more than 66% of tags was considered the taxonomic rank of a specific OTU. The bacterial diversity was analyzed by principal coordinate analysis.

## Transcriptome sequencing and gene identification

To identify the genes that contribute to converting protein into threonine, the transcriptome sequencing results of males fed different foods (sucrose, sucrose + YH, glucose, glucose + YH, fructose, and fructose + YH) were compared. For each group, five rectums were dissected for RNA extraction. In addition, five replicates were included for each group. In the next step, paired-end RNA-seq libraries were prepared by following Illumina's library construction protocol. The libraries were sequenced on an Illumina HiSeq2000 platform (Illumina, USA). FASTQ files of raw reads were produced and sorted by barcodes for further analysis. Prior to assembly, paired-end raw reads from each cDNA library were processed to remove adaptors, low-quality sequences (Q<20), and reads contaminated with microbes. The clean reads were de novo assembled to produce contigs. An index of the reference genome of *B. dorsalis* was built, and paired-end clean reads were mapped to the reference genome using HISAT2. 2.4 with '-rna-strandness RF' and other parameters set as a default (*Kim et al., 2015*). To evaluate transcript expression abundances, StringTie software was applied to calculate the normalized gene expression value FPKM (*Pertea et al., 2016*). Then, gene DE analysis was performed with DESeq2 software (*Love et al., 2014*). Genes/transcripts with a false discovery rate below 0.01 and absolute fold change ≥2 were considered DEGs/transcripts. Correlation analysis of the samples was performed by R. The correlation coefficient between two samples was calculated to evaluate similarity between samples. The closer the correlation coefficient is to 1, the higher the similarity between the two samples. To reveal the structure/relationship of the samples, PCA was performed with the R package gmodels. To further understand gene biological functions, pathway enrichment analysis was performed to identify the significantly enriched metabolic pathways or signal transduction pathways in DEGs compared with the whole genome. Moreover, the transcriptome of the normally reared (both sugar and protein were provided) male rectum at different developmental times (0 d, 3 d, 6 d, 9 d, and 12 d) was also sequenced and compared according to the above methods.

## Expression validation of the identified genes

qRT-PCR analysis was used to validate gene expression in the rectum, head, thorax, and abdomen of the males. Total RNA was extracted. Then, cDNA was synthesized with a One-Step gDNA Removal and cDNA Synthesis SuperMix Kit (TransGen Biotech, Beijing, China) using the extracted RNA. Then, a PerfectStarTM Green qPCR SuperMix Kit (TransGen Biotech, Beijing, China) was used to perform

quantitative real-time PCR to compare the gene expression levels. Gene-specific primers (*Supplementary file 9*) were designed on NCBI with primer blast. The *α-tubulin* and *actin* genes were used as reference genes (*Shen et al., 2010*). The PCR procedure was set according to the manufacturers' instructions. Five biological replicates were performed.

## RNA interference

dsRNA primers (*Supplementary file 9*) tailed with the T7 promoter sequence were designed using the CDSs(Coding DNA Sequence) of Sardh as templates. A MEGAscript RNAi Kit (Thermo Fisher Scientific, USA) was used to synthesize and purify dsRNA according to the manufacturer's instructions. The GFP gene (GenBank accession number: AHE38523) was used as the RNAi negative control. To knockdown the target gene in males, 0.5 µl (500 ng/µl) dsRNA was injected into the abdomen of 12 d old males. Flies injected with dsGFP(double strain RNA of green fluorescent protein) were prepared as a negative control. After 24 hr, the knockdown efficiency of the genes was checked with qRT-PCR following the method used for validating the expression of the gene above. Then, the threonine and glycine contents, sex pheromone abundance, and mating ability were measured and tested in flies in which Sardh was silenced.

## Data analysis

Statistical analysis methods used in the study were indicated in the figure legends. Differences were considered significant when p<0.05. All data were analyzed using the GraphPad Prism version 8, GraphPad Software, La Jolla, CA, USA, https://www.graphpad.com/.

## Acknowledgements

We are grateful to the national natural science foundation of China (No. 3212200346).

## Additional information

### Funding

| Funder | Grant reference number | Author |
|---|---|---|
| The national natural science foundation of China | 3212200346 | Daifeng Cheng |

The funders had no role in study design, data collection and interpretation, or the decision to submit the work for publication.

### Author contributions

Shiyu Gui, Formal analysis, Validation, Investigation, Visualization, Methodology; Boaz Yuval, Tobias Engl, Yongyue Lu, Writing - review and editing; Daifeng Cheng, Conceptualization, Resources, Data curation, Supervision, Funding acquisition, Validation, Methodology, Project administration, Writing - review and editing

### Author ORCIDs

Daifeng Cheng (iD) http://orcid.org/0000-0003-0918-5913

### Decision letter and Author response

Decision letter https://doi.org/10.7554/eLife.83469.sa1
Author response https://doi.org/10.7554/eLife.83469.sa2

## Additional files

### Supplementary files

• Supplementary file 1. OTU information of the 16S rRNA gene amplicons.
• Supplementary file 2. Alpha diversity analysis of the 16S rRNA gene amplicons.

• Supplementary file 3. Gene expression profiles in rectum of male fed with different type of food.

• Supplementary file 4. Expression profiles of the differential expression (DE) genes screened between yeast hydrolysate (YH)-fed and YH-deprived males.

• Supplementary file 5. Kyoto Encyclopedia of Genes and Genomes (KEGG) enrichment for the differential expression (DE) genes screened between yeast hydrolysate (YH)-fed and YH-deprived males.

• Supplementary file 6. Gene expression profiles in rectum of male at different development stages.

• Supplementary file 7. Expression profiles of the differential expression (DE) genes screened between different aged males.

• Supplementary file 8. Kyoto Encyclopedia of Genes and Genomes (KEGG) enrichment for the differential expression (DE) genes screened between different aged males.

• Supplementary file 9. Primers used in the study.

• MDAR checklist

## Data availability

All data needed to evaluate the conclusions in the paper are present in the paper and/or the Supplementary Materials. RNA-sequencing and 16S rRNA amplicon sequencing data have been deposited in the Genome Sequence Read Archive Database of the National Genomics Data Center (BioProject PRJCA010569, PRJCA010560 and PRJCA010555).

The following datasets were generated:

| Author(s) | Year | Dataset title | Dataset URL | Database and Identifier |
|---|---|---|---|---|
| Cheng D | 2022 | Post-teneral protein rectum transcriptome | https://ngdc.cncb. ac.cn/bioproject/ browse/PRJCA010569 | CNCB-NGDC, PRJCA010569 |
| Cheng D | 2022 | Rectum transcriptome at different development stage | https://ngdc.cncb. ac.cn/bioproject/ browse/PRJCA010560 | CNCB-NGDC, PRJCA010560 |
| Cheng D | 2022 | Effect of post-teneral protein on sex pheromone | https://ngdc.cncb. ac.cn/bioproject/ browse/PRJCA010555 | CNCB-NGDC, PRJCA010555 |

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
