## [Editor Report]

The manuscript describes the effects of dietary yeast and sugars on male Bactrocera dorsalis sex pheromone biosynthesis and other mating-related traits. This is an important study showing that yeast feeding stimulates the production of specific sex pheromones and promotes fly mating ability. The data are solid and will be of interest to the fields of chemical ecology and pest management.

---

## [Decision Letter]

**Decision letter after peer review:**

Thank you for submitting your article "Protein feeding mediates sex pheromone biosynthesis in an insect" for consideration by *eLife*. Your article has been reviewed by 2 peer reviewers, including Sonia Sen as the Reviewing Editor and Reviewer #1, and the evaluation has been overseen by K VijayRaghavan as the Senior Editor.

Essential revisions:

1. To strengthen the claim that rectal bacilli use the fly's dietary amino acid to make TMP/TTMP, we recommend that the authors demonstrate that in the absence of bacilli, the dietary amino acids do not make a difference to TMP/TTMP level or mating success.

2. Increase in levels of glycine and threonine: There is a possibility that these levels are elevated because the flies fed on YH and not because a specific threonine and glycine biosynthesis pathway was elevated. Could the authors please explicitly discuss this?

3. Figures: The figures could use considerable improvement. Please see the detailed reviews below for specifics.

4. Text: We request that the authors explain their experimental setups in more detail. Could they also please describe their results before they infer from them?

5. Please normalise the glycine/threonine content to tissue weight or total protein content rather than the sample volume.

In addition to this, please read the detailed reviews provided below and address them wherever possible.

*Reviewer #1 (Recommendations for the authors):*

The link between dietary protein and reproductive success is well documented in females of various species. Here, the authors investigate this link in male Oriental Fruitflies, which are a major agricultural pest.

In their previous work, the authors demonstrated that bacteria in the male rectum produce sex pheromones to attract females. In this study, the authors demonstrate that dietary protein triggers the expression of the gene, sarcosine dehydrogenase (sardh) in the male rectum. This, in turn, results in an increase in the levels of glycine and threonine, which are the precursors used by bacteria to produce the sex pheromones TMP and TTMP. Consequently, they demonstrate that protein feeding in males is necessary for mating success.

We found the manuscript interesting, and the claims generally well supported, particularly in the context of their previous work. Our recommendations are largely for clarity and presentation of data.

1. Figures: The figures could use considerable improvement. In some cases the graphs are too small to decipher the various conditions, occasionally the resolution is low, and 3D graphs are not ideal. In general, we recommend the use of schematics depicting the experiments and their conditions, and a final schematic to demonstrate their working model will be nice.

2. Text: It would help to have the experimental conditions better explained. For example, the considerations of the various sugar conditions are not really addressed in the text except in a single sentence. The interpretations from these different sugar conditions are also not really discussed, leaving one wondering why they were introduced in the first place. It would also help to have the experimental design briefly discussed in the Results section so that the results are easily interpretable by the reader without having to go to the methods.

3. The link between rectal microbial communities and TMP: The authors make the claim that yeast feeding does not affect bacterial communities. I'm not sure if this claim can be made based on the data, or at least this level of analysis of the data. I am not very familiar with community ecology analyses, but the indices used here are relative abundances of bacterial classes and overall richness. It's suggestive of their claim, but there is a possibility that species composition might be different in these conditions. If the authors agree, perhaps they could bring this point out and temper their claims.

4. The sentence in 107-108: "Yeast feeding only positively influenced glucose contents in males in the sucrose and fructose groups (Figure 2D)." The comparison seems to be being made between YH^+^ and YH- in each sugar group. The assessment to be made is whether the amount of glucose is affected by yeast feeding or not. So, should the comparison not be made across sugar groups? Glucose will naturally be higher in a glucose-fed fly, hence the apparent lack of difference between YH- and YH^+^.

5. It wasn't clear to me whether the authors were making the claim that Sardh levels were increased with age irrespective of the protein feeding. This would be interesting! The experimental setup of this section was particularly unclear to me.

6. The methods, for example, the bioinformatics pipelines, could be described better, particularly since this is a non-traditional model organism. Was the functional enrichment analysis done keeping the whole genome as the background? This won't help in determining the enrichment of genes within a specific tissue.

*Reviewer #2 (Recommendations for the authors):*

Gui et al. investigated the effects of dietary yeast and sugars on male Bactrocera dorsalis sex pheromone biosynthesis and other mating-related traits (rectum width, rectal glucose, threonine, glycine content, and mating ability). The authors first compared male B. dorsalis fed on diets with or without yeast hydrolase ("YH-supplemented" vs. "YH-deprived") and on three different sugars (glucose, fructose, and sucrose), showing that the sex pheromone compounds (particularly TMP) were detected in YH-supplemented male fly rectums but are absent in YH-deprived counterparts. The mating ability was also significantly lower in the YH-deprived males. Following the initial observations, the team conducted a series of experiments, including 16S amplicon seq, RNAseq, and gene knockdown by RNAi to identify the mechanism underlying these effects.

The manuscript is overall well-written, and the figures are nicely presented. Results demonstrating the importance of the sarcosine dehydrogenase (Sardh) gene in sex pheromone biosynthesis and mating are novel. However, I have reservations about some of the statements and data interpretation.

1. Previous work from the group suggests that the rectal bacteria can produce sex pheromones in male B. dorsalis (Ren, Ma et al. 2021). However, the involvement of bacteria in the dietary yeast effects on sex pheromone biosynthesis and mating is not clear in this study. The microbiome data suggest that the dietary yeast treatments did not affect the dominant rectal bacterium Bacillus abundance. The only indirect evidence was that rectal threonine and glycine appeared to be elevated in YH-fed flies, and Bacillus can use these amino acids to produce sex pheromones. In my opinion, the data present are not sufficient to support statements like "We show that male flies rely on rectal Bacillus to produce a complex sex pheromone. (L11-12)", "This study clearly links male feeding behavior with discrete patterns of gene expression that lead to pheromone production by rectal Bacillus… (L18-20)", "This is the first report that clarifies the molecular mechanism by which host feeding and rectal microbes co-regulate sex pheromone biosynthesis. (L173-174)". For these statements to be valid, one would need to use flies deprived of Bacillus or the microbiome to confirm that there is no difference between YH-fed and YH-deprived flies.

2. Some sections of the manuscript would benefit from clarification, for instance,

a. The authors reasoned that YH feeding would stimulate threonine and glycine synthesis pathways (L116-117) when they justified focusing on glycine/threonine metabolism DEGs. This is an interesting prediction but presumably, yeast can provide these amino acids directly as a protein source. Knowing how much threonine and glycine are available in the YH might be helpful.

b. The RNAseq data identified two DEGs, Sardh and AGXT2, associated with glycine or threonine synthesis. As shown in Figure 3E, Sardh can convert Sarcosine to glycine, but not much information was provided regarding where Sarcosine may come from. Does the YH provision Sarcosine to the fly? What is the alternative hypothesis for the Sardh upregulation by YH feeding if not?

3. Based on the fold changes and p values, the positive effect of YH feeding on rectal threonine level is more significant than on glycine (Figure 2E and 2F). Sardh upregulation can explain the elevated glycine level in YH-fed flies, but ltaE (that converts glycine to threonine) was not differentially expressed. The genetic basis for the elevated threonine seems unresolved and is worthy of more in-depth discussion.

4. Figure 1C shows that TTMP (blue arrow) is only detected in the glucose + YH treatment but not in the sucrose + YH or fructose + YH treatments. What are the possible explanations?

5. In Figure 5, Sardh knockdown males showed significantly decreased rectal threonine and glycine contents, but TTMP level was not significantly reduced. What are the possible explanations?

6. Please indicate the sample time (day 12?) of rectal glycine/threonine for figures 2 and 5.

7. For RNAseq analysis, the FPKM normalization method is generally not recommended for between-sample comparisons for DE analysis. Suggest using the median of ratios method in DESeq2 or TMM in EdgeR.

8. L203: "Our results indicate that the amount of sex pheromone produced is significantly affected by the type of sugar (Figure 2c)." – I think the authors were referring to Figure 1c, not 2c.

9. To measure rectal glycine/threonine content, the team collected 15 male recta per replicate of each treatment group. While rectal width did not differ between the YH-supplemented and YH-deprived flies, it's unclear whether the total protein content or mass was different between the groups. My concern is that the lower glycine/threonine level could be due to the lower amount of tissue collected from the YH-deprived flies. I also noticed that the scale of glycine content varied quite a bit among experiments (e.g., 1-2 μg/ml in Figure 2E, 0.4-0.6 μg/ml in Figure 4B, and 2-3 μg/ml in Figure 5C control group). Normalizing the glycine/threonine content to tissue weight or total protein content will be more appropriate than the sample volume.

---

## [Author Response]

Essential revisions:1. To strengthen the claim that rectal bacilli use the fly's dietary amino acid to make TMP/TTMP, we recommend that the authors demonstrate that in the absence of bacilli, the dietary amino acids do not make a difference to TMP/TTMP level or mating success.

Thanks for your recommendation. In previous study, we have found that flies can't synthesize sex pheromones when recta bacilli were removed by antibiotics treatment (Ren, Ma et al. 2021). Therefore, it is difficult to assess the effect of dietary amino acids on sex pheromones in the absence of bacilli (With or without dietary amino acids, there was no sex pheromone synthesis). To make the claim clearer, we have mentioned this in the manuscript. See line 79-84. We hope the explanation will meet with approval.

2. Increase in levels of glycine and threonine: There is a possibility that these levels are elevated because the flies fed on YH and not because a specific threonine and glycine biosynthesis pathway was elevated. Could the authors please explicitly discuss this?

Thanks for your comments. We agree with your opinion that glycine and threonine levels may be elevated because the flies fed on YH. We have discussed such possibility and the ways to make it clear in discussion. See line 196-199. We hope the correction will meet with approval.

3. Figures: The figures could use considerable improvement. Please see the detailed reviews below for specifics.

Thank you very much for you valuable suggestions. We have made corrections to the figures.

4. Text: We request that the authors explain their experimental setups in more detail. Could they also please describe their results before they infer from them?

Thanks for your suggestion. Experimental setups were described in more detail with more statements and schematics. And we have described the results in the result part. We hope the corrections will meet with approval.

5. Please normalise the glycine/threonine content to tissue weight or total protein content rather than the sample volume.

We have done this for both glycine/threonine contents and glucose contents.

In addition to this, please read the detailed reviews provided below and address them wherever possible.Reviewer #1 (Recommendations for the authors):The link between dietary protein and reproductive success is well documented in females of various species. Here, the authors investigate this link in male Oriental Fruitflies, which are a major agricultural pest.In their previous work, the authors demonstrated that bacteria in the male rectum produce sex pheromones to attract females. In this study, the authors demonstrate that dietary protein triggers the expression of the gene, sarcosine dehydrogenase (sardh) in the male rectum. This, in turn, results in an increase in the levels of glycine and threonine, which are the precursors used by bacteria to produce the sex pheromones TMP and TTMP. Consequently, they demonstrate that protein feeding in males is necessary for mating success.We found the manuscript interesting, and the claims generally well supported, particularly in the context of their previous work. Our recommendations are largely for clarity and presentation of data.

Thank you very much for your positive comments.

1. Figures: The figures could use considerable improvement. In some cases the graphs are too small to decipher the various conditions, occasionally the resolution is low, and 3D graphs are not ideal. In general, we recommend the use of schematics depicting the experiments and their conditions, and a final schematic to demonstrate their working model will be nice.

Thanks for your suggestion. We have improved the figures by increasing the resolution. And we have added schematics to demonstrate the working model. We hope the corrections will meet with approval.

2. Text: It would help to have the experimental conditions better explained. For example, the considerations of the various sugar conditions are not really addressed in the text except in a single sentence. The interpretations from these different sugar conditions are also not really discussed, leaving one wondering why they were introduced in the first place. It would also help to have the experimental design briefly discussed in the Results section so that the results are easily interpretable by the reader without having to go to the methods.

We are sorry for the confusion caused. We have described the experimental conditions in detail in the result and method part. See line 68-72, line 242-243. We hope the corrections will meet with approval.

3. The link between rectal microbial communities and TMP: The authors make the claim that yeast feeding does not affect bacterial communities. I'm not sure if this claim can be made based on the data, or at least this level of analysis of the data. I am not very familiar with community ecology analyses, but the indices used here are relative abundances of bacterial classes and overall richness. It's suggestive of their claim, but there is a possibility that species composition might be different in these conditions. If the authors agree, perhaps they could bring this point out and temper their claims.

We agree with you very much. In order to clarify this issue, we have rewritten this part. We have only highlighted the effect of protein supplementation on microbes at the class level. And we have tempered our claims by saying that the results indicate that protein intake may not affect the abundance of Bacillus synthesizing pheromones in the rectum and sex pheromone loss in YH-deprived males may not be associated with Bacillus. See line 88-97. Thanks again for your important suggestions. We hope the revision will meet with approval.

4. The sentence in 107-108: "Yeast feeding only positively influenced glucose contents in males in the sucrose and fructose groups (Figure 2D)." The comparison seems to be being made between YH^+^ and YH- in each sugar group. The assessment to be made is whether the amount of glucose is affected by yeast feeding or not. So, should the comparison not be made across sugar groups? Glucose will naturally be higher in a glucose-fed fly, hence the apparent lack of difference between YH- and YH^+^.

Thanks for your comments. In our study, we found that protein supplementation can significantly affect the synthesis of sex pheromones. Therefore, we wanted to further confirm which part (Bacillus, glucose and amino acids) in sex pheromone synthesis was affected by protein supplementation. In the issue you have mentioned, we want to clarify whether protein supplementation affects sex pheromone synthesis by influencing the rectal glucose contents. We are sorry that we didn't make this issue clear. We have restated the purpose of such experiments in the manuscript. See line 98-104. We hope the explanation will meet with approval.

5. It wasn't clear to me whether the authors were making the claim that Sardh levels were increased with age irrespective of the protein feeding. This would be interesting! The experimental setup of this section was particularly unclear to me.

We are sorry for the unclear description of the experimental setup. In our previous study, we have found that sex pheromones can only be produced 9 days after emergence in normally reared (sugar and protein were both provided) male B. dorsalis. Thus, we infer the reason may be that Sardh levels were increased in mature stage to synthesize amino acid for sex pheromone synthesis. And we can set such experiment to verify the potential role of Sardh in influencing sex pheromone synthesis. We have made correction in the results and method part. Sorry again for the confusion caused. We hope the corrections and explanation will meet with approval.

6. The methods, for example, the bioinformatics pipelines, could be described better, particularly since this is a non-traditional model organism. Was the functional enrichment analysis done keeping the whole genome as the background? This won't help in determining the enrichment of genes within a specific tissue.

Thanks for your comments. Before we did pathway enrichment, we already knew that threonine/glycine contents were affected by protein supplementation. Therefore, the purpose of pathway enrichment analysis is to further screen genes involved in synthesizing threonine/glycine. Doing functional enrichment analysis with the whole genome as the background is commonly used in many literatures. We certainly agree with you that using genes expressed in the rectum as background will make the results more credible. And we re-did the enrichment analysis with the method you recommended, and the results also showed that the glycine and threonine metabolism pathway was significantly enriched. Thanks again for your valuable comments. We hope our explanation will meet with approval.

Reviewer #2 (Recommendations for the authors):Gui et al. investigated the effects of dietary yeast and sugars on male Bactrocera dorsalis sex pheromone biosynthesis and other mating-related traits (rectum width, rectal glucose, threonine, glycine content, and mating ability). The authors first compared male B. dorsalis fed on diets with or without yeast hydrolase ("YH-supplemented" vs. "YH-deprived") and on three different sugars (glucose, fructose, and sucrose), showing that the sex pheromone compounds (particularly TMP) were detected in YH-supplemented male fly rectums but are absent in YH-deprived counterparts. The mating ability was also significantly lower in the YH-deprived males. Following the initial observations, the team conducted a series of experiments, including 16S amplicon seq, RNAseq, and gene knockdown by RNAi to identify the mechanism underlying these effects.The manuscript is overall well-written, and the figures are nicely presented. Results demonstrating the importance of the sarcosine dehydrogenase (Sardh) gene in sex pheromone biosynthesis and mating are novel. However, I have reservations about some of the statements and data interpretation.

Thanks for your positive comments.

1. Previous work from the group suggests that the rectal bacteria can produce sex pheromones in male B. dorsalis (Ren, Ma et al. 2021). However, the involvement of bacteria in the dietary yeast effects on sex pheromone biosynthesis and mating is not clear in this study. The microbiome data suggest that the dietary yeast treatments did not affect the dominant rectal bacterium Bacillus abundance. The only indirect evidence was that rectal threonine and glycine appeared to be elevated in YH-fed flies, and Bacillus can use these amino acids to produce sex pheromones. In my opinion, the data present are not sufficient to support statements like "We show that male flies rely on rectal Bacillus to produce a complex sex pheromone. (L11-12)", "This study clearly links male feeding behavior with discrete patterns of gene expression that lead to pheromone production by rectal Bacillus… (L18-20)", "This is the first report that clarifies the molecular mechanism by which host feeding and rectal microbes co-regulate sex pheromone biosynthesis. (L173-174)". For these statements to be valid, one would need to use flies deprived of Bacillus or the microbiome to confirm that there is no difference between YH-fed and YH-deprived flies.

Thanks for your important comments. Although we have shown in previous studies that Bacillus can synthesize sex pheromones, these conclusions are not part of this study. In order to be more precise, we have rephrased or deleted the statements in the manuscript. Thanks again for your valuable comments. We hope the corrections will meet with approval.

2. Some sections of the manuscript would benefit from clarification, for instance,a. The authors reasoned that YH feeding would stimulate threonine and glycine synthesis pathways (L116-117) when they justified focusing on glycine/threonine metabolism DEGs. This is an interesting prediction but presumably, yeast can provide these amino acids directly as a protein source. Knowing how much threonine and glycine are available in the YH might be helpful.

Thanks for your important comments. We agree with your opinion that glycine and threonine levels may be elevated because the flies fed on YH. And the editor also points out that we need to discuss such possibility. However, our results at least indicate that glycine/threonine metabolism pathway is one of the factors regulating glycine/threonine and sex pheromone synthesis in rectum. Taking the editor's advice into consideration, we have discussed such possibility and the ways to make it clear in discussion. See line 196-199. We hope the corrections will meet with approval.

b. The RNAseq data identified two DEGs, Sardh and AGXT2, associated with glycine or threonine synthesis. As shown in Figure 3E, Sardh can convert Sarcosine to glycine, but not much information was provided regarding where Sarcosine may come from. Does the YH provision Sarcosine to the fly? What is the alternative hypothesis for the Sardh upregulation by YH feeding if not?

Thank you very much for your important comments. Indeed, the issues you have raised are critical for understanding the mechanism that protein supplementary regulated sex pheromone synthesis. We need a series of experiments to clarify these issues, which is almost a whole other topic. However, we agree with you very much. Considering the suggestions of the editor, we have carried out an in-depth discussion in the discussion part, and proposed further research directions. See line 216-230. Thanks again for your comments. We hope our corrections will meet with approval.

3. Based on the fold changes and p values, the positive effect of YH feeding on rectal threonine level is more significant than on glycine (Figure 2E and 2F). Sardh upregulation can explain the elevated glycine level in YH-fed flies, but ltaE (that converts glycine to threonine) was not differentially expressed. The genetic basis for the elevated threonine seems unresolved and is worthy of more in-depth discussion.

We agree with your opinion. We inferred elevated threonine may come from two sources. One is provided directly by YH, another one is converted by other genes. However, we need more date to confirm such speculation. According to your suggestions and the editor’s, we have mentioned this in the discussion part. See line 216-230. We hope the correction will meet with approval.

4. Figure 1C shows that TTMP (blue arrow) is only detected in the glucose + YH treatment but not in the sucrose + YH or fructose + YH treatments. What are the possible explanations?

We are very sorry for our unclear description. TTMP has been detected in the glucose + YH, sucrose + YH and fructose + YH treatments. In order to show the results clearly, we added some extra marks in the Figure. Sorry again for our negligence. We hope the correction will meet with approval.

5. In Figure 5, Sardh knockdown males showed significantly decreased rectal threonine and glycine contents, but TTMP level was not significantly reduced. What are the possible explanations?

Thanks for your important comments. In the pyrazine synthesis pathway of Bacillus, two molecules of glucose can be converted into TTMP, while one molecule of glucose and one molecule of threonine (or glycine) can be converted into TMP (Zhang et al., 2019 Applied and Environmental Microbiology). Therefore, we speculate that the reason why TTMP level is not affected is that glucose content in the rectum is not regulated by Sardh. Thanks for your question. We are sorry that we didn’t discuss this in the manuscript. In the revised manuscript, we have added the explanations in discussion. See line 216-230. We hope the revisions will meet with approval.

6. Please indicate the sample time (day 12?) of rectal glycine/threonine for figures 2 and 5.

Thank you very much for your reminding. The flies used were mature males (12-day-old). We have mentioned this in the manuscript.

7. For RNAseq analysis, the FPKM normalization method is generally not recommended for between-sample comparisons for DE analysis. Suggest using the median of ratios method in DESeq2 or TMM in EdgeR.

Thanks for your suggestions. We have found that both FPKM and TPM are used for quantitative analysis in transcriptome in many literatures. Although the quantitative methods are different, the results obtained are highly consistent. And FPKM is still the main reference transcriptome quantitative method at present. We analyzed the date again with the method you recommended and found it was highly consistent with our previous results. So we think there is no problem in using FPKM normalization method. Thanks again for your recommendation. We hope our explanation will meet with approval.

8. L203: "Our results indicate that the amount of sex pheromone produced is significantly affected by the type of sugar (Figure 2c)." – I think the authors were referring to Figure 1c, not 2c.

We are sorry for our mistake. We have corrected it.

9. To measure rectal glycine/threonine content, the team collected 15 male recta per replicate of each treatment group. While rectal width did not differ between the YH-supplemented and YH-deprived flies, it's unclear whether the total protein content or mass was different between the groups. My concern is that the lower glycine/threonine level could be due to the lower amount of tissue collected from the YH-deprived flies. I also noticed that the scale of glycine content varied quite a bit among experiments (e.g., 1-2 μg/ml in Figure 2E, 0.4-0.6 μg/ml in Figure 4B, and 2-3 μg/ml in Figure 5C control group). Normalizing the glycine/threonine content to tissue weight or total protein content will be more appropriate than the sample volume.

Thank you very much for your valuable suggestions. We have normalized both glycine/threonine content and glucose content to tissue weight in the revised manuscript.